# The Impact of Resolution of Inflammation on Tumor Microenvironment: Exploring New Ways to Control Cancer Progression

**DOI:** 10.3390/cancers14143333

**Published:** 2022-07-08

**Authors:** Federica Liotti, Maria Marotta, Rosa Marina Melillo, Nella Prevete

**Affiliations:** 1Department of Molecular Medicine and Medical Biotechnology, University of Naples Federico II, 80131 Naples, Italy; federica.liotti@unina.it (F.L.); maria.m_1994@libero.it (M.M.); 2Institute of Experimental Endocrinology and Oncology (IEOS), CNR, 80131 Naples, Italy; 3Department of Translational Medical Sciences, University of Naples Federico II, 80131 Naples, Italy

**Keywords:** resolution of inflammation, specialized pro-resolving mediators, tumor microenvironment

## Abstract

**Simple Summary:**

The evolution of cancer is strongly influenced by the context in which tumor cells develop and grow, known as the tumor microenvironment (TME). The TME is constituted of a set of cells with different natures, which can produce various factors or interact with cancer cells, thus favoring or inhibiting cancer growth. Specific factors with the ability to shape the TME, in order to create an unfavorable context for tumor cells, are the Specialized Pro-resolving Mediators (SPMs). SPMs are small lipid molecules derived from ω-3 and ω-6 fatty acids, exerting the physiologic role of dampening the inflammatory responses and helping tissues to regain their homeostasis after insults. Here, we present the knowledge relative to the action of SPMs on each component of the TME and its effects on tumor growth and progression. These summarized findings highlight novel potential strategies to manage cancer progression.

**Abstract:**

Non-resolving inflammation is an enabling feature of cancer. A novel super-family of lipid mediators termed Specialized Pro-resolving Mediators (SPMs) have a role as bioactive molecules mediating the resolution of inflammation in cancer biology. SPMs are derived from ω-3 and ω-6 polyunsaturated fatty acids through the activity of lipoxygenases. SPMs have been described to directly modulate cancer progression by interfering with the epithelial to mesenchymal transition and invasion of cancer cells. SPMs have also been demonstrated to act on several components of the tumor microenvironment (TME). Consistently with their natural immunomodulatory and anti-inflammatory properties, SPMs are able to reprogram macrophages to favor phagocytosis of cell debris, which are an important source of pro-inflammatory and pro-angiogenic signals; sustain a direct cytotoxic immune response against cancer cells; stimulate neutrophils anti-tumor activities; and inhibit the development of regulatory T and B cells, thus indirectly leading to enhanced anti-tumor immunity. Furthermore, the resolution pathways exert crucial anti-angiogenic functions in lung, liver, and gastrointestinal cancers, and inhibit cancer-associated fibroblast differentiation and functions in hepatocellular carcinoma and pancreatic cancer. The present review will be focused on the potential protective effects of resolution pathways against cancer, exerted by modulating different components of the TME.

## 1. Resolution of Inflammation

Chronic inflammation, together with genome instability, are now considered enabling features of cancer, fostering the other established hallmark functions (i.e., sustained proliferation and resistance to cell death, increased angiogenesis, and invasion capability, etc.) [1]. Consistently, several studies have demonstrated that inflammation intervenes in tumor initiation, growth, and progression in different cancer models [2].

In recent years, various mechanisms underlying the onset of a chronic inflammatory process have been defined. In particular, it is now clear that inflammation intensity and duration are the result of the balance of two active processes: the first is the activation of inflammatory response with production of pro-inflammatory mediators (e.g., eicosanoids and cytokines) and recruitment of innate immune cells; the second is its resolution process mediated by different actively produced mediators, whose goal is to dampen the inflammatory response and allow the return to tissue homeostasis [3].

While inflammation is characterized by the production of inflammatory mediators facilitating leukocyte recruitment and activation, the resolution phase stands out for pro-inflammatory mediator degradation and/or reduced production. When the concentrations of pro-inflammatory factors reach a plateau, neutrophils and the other leukocytes infiltrating the inflamed site initiate a switch in the mediators produced, from pro-inflammatory to pro-resolutive, in order to counteract further recruitment of inflammatory neutrophils and to favor non-inflammatory monocytes. An example of this fine adjustment is the change in the lipidic mediators present in the exudates: in the first phase of inflammation, prostaglandins and leukotrienes are abundant, while during the resolution process different kinds of lipidic autacoids can be detected (e.g., lipoxins and resolvins). Pro-resolving mediators not only limit the inflammatory response, but also contribute to tissue homeostasis, causing the clearance of apoptotic neutrophils (aka, efferocytosis), sustaining several steps of tissue repair, and limiting fibrosis and scar formation [4].

Different kinds of mediators can actively sustain the resolution response [4]. Lipidic mediators intervene in inflammation resolution: these autacoids are known as Specialized Pro-resolving Mediators [lipoxins (LX), resolvins (Rv), protectins (PD), and maresins (MaR)] [5]. Furthermore, annexin A1 (AnxA1), together with other proteins (e.g., adrenocorticotropic hormone, chemerin peptides, and galectin-1) can sustain the resolution program [4,6]. Interestingly, gaseous mediators (nitric oxide, hydrogen sulfide, and carbon monoxide), adenosine, and neuromodulators such as acetylcholine could also exert anti-inflammatory and pro-resolving functions [7,8].

An insufficient resolution response can lead to chronic inflammation, exactly as in the persistent activation of a pro-inflammatory program [9]. However, while the study of the resolution of inflammation in acute inflammatory response has been widely dissected [10], the investigation of its role in chronic inflammatory processes is more recent and therefore less detailed [11].

## 2. Specialized Pro-Resolving Mediators and Cancer

Specialized Pro-resolving Mediators (SPMs) are bioactive lipid molecules produced from ω-3 (docosahexaenoic (DHA) and eicosapentaenoic (EPA) acids) and ω-6 (arachidonic acid (AA)) polyunsaturated fatty acids (PUFA) thanks to the activity of specific enzymes (lipoxygenases (ALOX5, ALOX15)). Lipoxins (LXA4, LXB4) are SPMs derived from the ω-6 AA; E-series Resolvins (RvEs) from the ω-3 EPA; D-series Resolvins (RvDs), Protectins (PD), and Maresins (MaR) from the ω-3 DHA [12]. These molecules exert their action by interacting with G protein coupled receptors (e.g., GPR18, GPR32, GPR37, ChemR23, FPR2, BLT1) [13,14].

FPR2, aka ALX/FPR2, contributes to the resolution of inflammation by binding LXA4 [15,16,17], RvD1, and RvD3 [18,19]. RvD1 and RvD3 have also been described as binding GPR32 [16,17], together with RvD5 [14]. ChemR23 was identified as the high affinity receptor for RvE1 [20]. RvE1, in addition to ChemR23, has also been described as binding with low affinity the BLT1 receptor [21]. ChemR23 also recognizes RvE2 [22]. Merlin and coll. recently published that they were not able to observe ChemR23 activation upon RvE1 stimulation of HEK293 cells [23], thus suggesting that more in depth studies are needed to define the pairs of receptor–ligands or to describe context-dependent effects of these receptor(s). GPR18, an already known receptor for cannabinoids [24], also recognizes RvD2 [25]. GPR37 is a candidate receptor for NPD1/PD1 [26], together with BLT1 [23]. More recently, other studies have shown that MaR1 is able to bind and activate the LGR6 receptor [27] and that N-3 docosapentaenoic acid-derived resolvin D5 (RvD5_n-3 DPA_) can activate resolution responses through the engagement of GPR101 receptor [28].

Due to their crucial role in the regulation of inflammatory responses, SPM deficiency has been linked to several pathologic conditions in which an inflammatory microenvironment is established and is the cause of disease, including cancer [29].

The strongest evidence of the crucial role of SPMs in the modulation of cancer initiation and progression comes from the studies of Serhan et al., who demonstrated that the protective role of aspirin against some cancer types could be ascribed to its ability to mediate COX2 acetylation. This COX2 modification is responsible for the production of the SPM epimers (aka, aspirin-triggered (AT) lipoxins/epi-lipoxins and aspirin-triggered (AT) resolvins/epi-resolvins) [12]. These effects are typical of aspirin and not common to other nonsteroidal anti-inflammatory drugs (NSAIDs); consistently, aspirin triggers beneficial effects on tumorigenesis that have not been observed with NSAIDs [30,31]. 

Although the study of SPM functions in cancer is new and the data produced are still limited, some evidence in humans supports a tumor suppressor function of SPMs [29,32,33,34].

The cancer types in which a protective role of SPMs has been described are oral, gastric, colon, pancreatic, liver, lung cancers, melanoma, papilloma, and non-solid tumors such as leukemia [35].

In recent years, attention has been paid to the mechanisms by which SPMs could affect cancer progression. Obviously, several of these activities have been linked to the anti-inflammatory potential of SPMs and their impact on different components of the tumor microenvironment (TME), but the direct effect of SPMs on cancer cells survival, epithelial to mesenchymal transition (EMT), and invasion also deserves to be mentioned [9,34,36,37,38,39,40,41,42] (Table 1).

The present review will be focused on the lipid Specialized Pro-resolving Mediators and their role in cancer progression, pointing a magnifying glass at their effects on tumor stroma. In the following paragraphs, we will describe the most important reports present in the literature related to the effects of SPMs on the three component of the TME: the immune cell infiltrate, the vascular bed, and the mesenchymal component of tumor stroma.

## 3. Effects of SPMs on the Immune Cell Compartment Infiltrating Tumor Microenvironment

Tumor-infiltrating immune cells play a key role in the evolution of cancer. Typically, immune cells should recognize and eliminate cancer cells [56]. Despite this, cancer cells continue to grow and expand, due to several escape mechanisms [57]. Furthermore, several immune cell classes can orchestrate cancer growth, by producing mediators directly sustaining cancer progression [58].

The most abundant immune cell population infiltrating tumors are the macrophages, dendritic cells (DCs), myeloid-derived suppressor cells (MDSCs), T cells, mast cells, and natural killer (NK) cells. Some of them have been clearly associated to anti-tumor immune responses: lymphoid cells, including natural killer (NK) cells, CD8^+^ T lymphocytes, CD4^+^ helper T (Th) cells, pro-inflammatory macrophages (M1), and DCs. Moreover, MDSCs and regulatory T (Treg) cells have been shown to exert a pro-tumorigenic action [59,60].

In the context of anti-tumor immune response, a primary regulatory role has to be ascribed to immune checkpoint (IC) molecules (e.g., Programmed Cell Death-1 (PD-1), Programmed death ligand 1 (PD-L1), Cytotoxic T-Lymphocyte Antigen 4 (CTLA-4), etc.). Due to its ability to dampen anti-tumor immune response, immune checkpoint blockade (ICB) is today considered, among the various types of immunotherapy, the pivotal approach [61]. It is now clear that a continuous influence of IC on TME, and vice versa, exists. Thus, it would also be simplistic not to consider, in the context of immune response to cancer, the possibility that resolution of inflammation could affect immune-checkpoint expression and functions.

In this scenario, an RNA-seq analysis of patients with head and neck squamous cell carcinoma showed that a higher score for RvD metabolism-associated markers directly correlates with a better clinical outcome. A high RvD score also associates to a signature suggestive of enhanced anti-tumor immunity, as demonstrated by its association to genes involved in the cytotoxic activity of immune cells (e.g., granzymes and perforin). Finally, a high RvD score is presumably linked to increased responsivity to immune-checkpoint inhibitors (ICI), as demonstrated by its significant correlation with the expression of four immune-checkpoint inhibitor targets (CD274, CTLA4, LAG3, and PDCD1) [62].

Consistently, preliminary data from Gartung et al. reported that, in a murine model of head and neck cancer, resolvins and immune checkpoint inhibitors (e.g., anti-PD-1) act synergistically, by suppressing tumor growth [63]. Furthermore, in a model of colorectal carcinoma (CRC), administration of AT-SPM regulated macrophage trafficking by stimulating the clearance of apoptotic cells and reduced the expression of the immunosuppressive receptor PD-1 in both macrophages and CD8^+^ T cells [64]. Similarly, SPM LXA4 was reported to reduce the expression of PD-L1 in a Kaposi’s sarcoma-associated herpesvirus (KSHV)-induced model [65].

### 3.1. SPM Effects on Innate Immune Cell Compartment

Due to their natural role in inhibiting the inflammatory response, most of the studies present in the literature, including those related to cancer, have been focused on the effects of SPMs in modulating the functions of innate immune cells (Table 1).

The most abundant innate immune cells infiltrating tumors are neutrophils and macrophages, which sometimes act against cancer cells through cytotoxic and phagocytic activities, while in other contexts they sustain tumor growth through the production of mediators hijacked by cancer cells to favor their spread [66]. 

Here, we will summarize the knowledge regarding SPM effects on neutrophils and macrophages. Furthermore, a paragraph will be dedicated to DCs, the innate immune cell population essential for the induction of an adaptive immune response.

#### 3.1.1. Neutrophils

While several research programs have highlighted the role of macrophages in cancer progression [67], less information is available concerning the function of neutrophils in the context of the tumor microenvironment [46]. To date, most reports have sustained the pro-tumorigenic role of neutrophils, and more generally of polymorphonuclear (PMN) cells [46,48], while others have demonstrated that PMN are able to inhibit cancer progression [68,69,70,71]. The literature suggests that the different functions of neutrophils in supporting or inhibiting cancer depends on the tumor stage and the cancer tissue [71].

Nevertheless, neutrophils are targets of resolution molecules, as demonstrated by the evidence that BLT1 receptor expressed on neutrophil surface is crucial in the regulation of their mitochondrial functions. In more detail, the RvE1-BLT1 signal mediates the activation of apoptotic responses through the induction of reactive oxygen species (ROS) and subsequent activation of caspases, and through the inhibition of ERK and Akt anti-apoptotic signals [72]. Furthermore, GPR18 knockout in mice displayed reduced neutrophil infiltration at the infection site and reduced tissue recovery following injuries [25].

The few studies regarding the effects of pro-resolving mediators on cancer-associated neutrophils were directed to the definition of the effects of SPMs on neutrophil phenotype and functions and on the impact of these on cancer progression.

Mattoscio and coll. demonstrated that RvD1 reduces the growth and proliferation of tumors in vivo using a human papilloma virus (HPV) tumorigenesis model. This effect is mediated by the ability of RvD1 to change the phenotype of neutrophils in a “pro-resolving anti-cancer phenotype”, characterized by increased anti-tumor properties. In addition, RvD1 stimulated PMN to secrete increased levels of chemoattractants for monocytes able to inhibit tumor growth in vivo [46].

Vannitamby et al. focused their studies on the effects of AT-RvD1 on neutrophils in a mouse model of lung cancer. AT-RvD1 is produced by aspirin-acetylated COX2 and 5-lipoxygenase (ALOX5) enzymes [48]. The authors found that increased neutrophil infiltration in the TME of lung adenocarcinoma patients correlated with a low level of expression of ALOX5 enzyme. Based upon this observation, they investigated the possibility that AT-RvD1 could exert a therapeutic activity in a mice model of lung adenocarcinoma, demonstrating that AT-RvD1 treatment significantly reduced lung adenocarcinoma growth in vivo by reducing the infiltration of neutrophil, which is responsible for lymphocyte activity suppression [48].

#### 3.1.2. Macrophages

Macrophages represent probably the most important target cell of SPMs. In the physiology of inflammation resolution, SPMs sustain the recruitment of the macrophages responsible for the non-inflammatory efferocytosis of apoptotic neutrophils [73,74].

The classic view of macrophage biology classifies them as M1, with pro-inflammatory, microbicidal, and anti-tumor functions; or as M2, displaying anti-inflammatory, immunosuppressive, and pro-tumorigenic properties [75]. However, this polarization model shows some limitations: it reflects well the characteristics of macrophages induced to polarize in vitro by specific cytokines and mediators, but it does not describe well the characteristics of macrophages present in tissues [76]. In the cancer context, tumor associated macrophages (TAM) display phenotypes that are a continuum between the two phenotypes, being continuously conditioned by the TME [76]. For example, the most abundant phenotype of macrophages in TME is M2, but these cells can exert a more efficient phagocytosis of cancer cells than M1 cells, thus potentially favoring an anti-tumor response [29].

A characterization of the lipid mediators produced by the two different classes of macrophages demonstrated that M1 produce pro-inflammatory lipid mediators, such as prostaglandins and leukotrienes, while M2 macrophages produce SPMs. This suggests that the different classes of lipid autacoids could be useful in re-shaping the macrophage phenotype to activate a protective anti-tumor response [29]. A critical role of resolution molecules in macrophage function regulation was also revealed in studies on SPM receptors knock-out: (i) macrophages deleted for GPR32 become unresponsive to RvD1 and lost the ability to polarize toward a pro-resolutive phenotype [19]; (ii) GPR18 knock-out in macrophages reduced their ability to phagocyte debris and dead cells [25]; (iii) ChemR23 blocking antibody reduced the phagocytic activity of RvE1-stimulated macrophages in acute and chronic inflammatory models [77,78]; and (iv) mice knocked-out for GPR37 displayed a reduced macrophage phagocytic activity in a model of inflammatory pain. [26].

Physiologically, SPMs increase macrophage survival [79], sustain the phagocytic activity of macrophages against microbes and apoptotic neutrophils [16,80], reduce the secretion of pro-inflammatory cytokines via inhibition of NF-kB [29,80], and, in the meantime, increase the production of anti-inflammatory cytokines [22,81]. 

Although it would be expected that the SPM-sustained changes of macrophage phenotype towards a M2 type would favor neoplastic progression, the available literature points to a protective anti-tumor effect of SPM-conditioned TAM [29,54]. The analysis of the effects of SPMs on TAM supports this evidence and demonstrates that the scenario is more complex than expected. Indeed, SPMs sustain changes in TAM that place them in between the M1 and M2 phenotypes [82,83,84].

Several research groups have investigated the role of SPMs in conditioning macrophage phenotype in the cancer context [29]. ATL-1, a synthetic analogue of LXA4, alters TAM profile by decreasing M2 surface markers, triggering ROS production, and increasing the cytotoxic and decreasing the anti-apoptotic properties of TAM against melanoma cancer cells. Consistently, ATL-1 inhibited cancer progression in a murine model by affecting macrophage phenotype [29]. Similarly, RvD1 and RvD2 inhibited cancer cell proliferation by affecting macrophage polarization in a prostate cancer model [47].

A very interesting research line has focused on the effects of SPMs on macrophage efferocytosis of chemotherapies-induced tumor cell debris. Debris of tumor cells have been demonstrated to be produced following several types of cancer therapies, including radiation and chemotherapy, and to sustain tumor engraftment, growth, and metastasis by activating a pro-inflammatory response [85,86]. Furthermore, tumor cell debris affect M1 TAM, inducing an immunosuppressive response, and thus limiting anti-cancer immunity [87,88].

The involvement of lipidic autacoids in the regulation of inflammatory response associated to cancer cell debris is now clear, since it has been recognized that they stimulate tumor growth by sustaining the production of the pro-inflammatory autacoid prostaglandin E2 (PGE2) in the TME [89]. On the other hand, in several tumor types, SPMs RvD1, RvD2, and RvE1 block the ability of tumor cell debris to sustain cancer progression via the activation of macrophage phagocytosis and clearance of cell debris [44]. SPMs sustained polarization of macrophages in these models is also characterized by a reduced production of pro-inflammatory cytokines [44]. Furthermore, the anti-tumor effect of aspirin was linked to its ability to induce the production of SPMs, enhance the macrophage phagocytosis of tumor cell debris, and reduce macrophage production of inflammatory mediators [45].

#### 3.1.3. Dendritic Cells

This cell type represents a crucial link between innate immunity and the activation of the adaptive response, due to its ability to professionally present antigens to lymphocytes [90]. DCs express receptors to SPMs, thus representing a target of these molecules [91]. To date, evidence has been produced demonstrating that SPMs could affect DC maturation and functions, with the aim of reducing inflammation and allowing tissue restitution. SPMs reduce the DC migration and production of pro-inflammatory cytokines [92,93]. Furthermore, RvD1 reduces the levels of Major Histocompatibility Complex II (MHC II) and costimulatory molecules on DCs [94]. As an example, RvE1 treatment blocks DCs in an immature state [95] and prevents their migration to draining lymph-nodes, thus resulting in a reduced T cell response [96,97]. All these evidence has been produced in disease models in which SPMs work in order to facilitate tissue homeostasis (e.g., infection, autoimmune diseases, etc.), but currently no data have been published on SPM effects on DCs in a cancer context, where the separation between tissue homeostasis and tumor growth is less clear.

### 3.2. Effects of SPMs on the Adaptive Immune Cell Compartment of the TME

Very few studies are available on the effects of SPMs on the adaptive immune branch of the immune system, and even less in the specific context of cancer, although the potential immunomodulatory effects of SPMs on adaptive immune response could be of great interest [98].

#### 3.2.1. Effector T and B lymphocytes

Since the activation of T or B lymphocytes is the goal of the adaptive immune response, an active area of study focuses on SPM activity with T and B cells. The effector functions of the different T and B cell subsets are strictly linked to the characteristics of the produced cytokines [99]. 

The most plastic subset of human cells is probably represented by the CD4^+^ T lymphocytes: the effector functions against different classes of pathogens are assigned to T helper (Th)1, Th2, and Th17 subsets; follicular helper T (Thf) cells assist B lymphocytes in their maturation, and Treg cells, instead, control self-tolerance [100]. Chiurchiù and coll. investigated the effects of SPMs on T CD4^+^ lymphocytes, demonstrating that RvD1, RvD2, and MaR1 are able to modulate T cell differentiation programs suppressing Th1 and Th17 T cell subtypes, as well as their ability to produce pro-inflammatory cytokines and to increase, in the meantime, the production of Treg cells [101]. Similarly, the inhibition of cytokine production mediated by SPMs on CD4^+^ Th1 and Th17 T cells has also been described for PD1, RvD3, LXA4, and LXB4 [102,103]. This inhibitory effect of SPMs on cytokine production has a great impact on pro-inflammatory T cell subsets (i.e., Th1 and Th17) but a less important effect on the Th2 phenotype. An exception was reported for RvE1, which has been described to dampen the Th2 T cell response in a model of asthma [104,105]. 

Both in vitro and in vivo data support the hypothesis that SPMs are instead able to augment Treg generation, consistently with their ability to produce immunomodulatory factors and to exert pro-resolving functions [101,106,107]. Since more data are available on the effects of SPMs on regulatory subsets of lymphocytes, a specific paragraph will be dedicated to these.

Unfortunately, to date, no direct and structured data have been published on the effects of SPMs on the cytotoxic response of CD8^+^ T lymphocytes, although they represent the key to directly killing cancer cells. However, in support of a role of SPMs in sustaining an anti-tumor cytotoxic response, it has been observed that RvD1 and its precursor (DHA) affect natural killer (NK) cells, the innate immune counterpart of CD8^+^ T cells, by preventing their death and sustaining their tumoricidal activity [38]. Consistently, it has been demonstrated that LXA4 is able to improve the anti-tumor activities of CD8^+^ T cells indirectly, by reducing regulatory subsets of T and B cells [53]. Furthermore, it has been described that GPR18 is expressed in CD8^+^ T lymphocytes and regulates their development and homing in epithelia [108], suggesting the crucial importance of resolution responses in tumor immune response and immunotherapy [108]. Interestingly, it has been demonstrated that BLT1 regulates anti-tumor immune responses by favoring CD8^+^ T cell migration in melanoma [109] and the cervical cancer murine microenvironment [110].

Some authors have also addressed the role of inflammation resolution on B cell compartment. SPMs have been described as potent modulators of B cell functions [111]: RvD1 increases IgM and IgG and decreases IL-6 production from B cells, as well as favoring the differentiation of B cells in antibody secreting cells [112]. Several other reports have pointed to a role of SPMs in sustaining antibody production from B cells [113,114]. LXA4 has been described to be able to accelerate B cell migration to the spleen, increasing the response to antigens [115]. 

These data perfectly fit into the scenario of SPMs as negative modulators of inflammatory response when acting on the adaptive immune compartment, although no structured data are available in cancer context nor a clear definition of potential pro- or anti-tumor effects.

#### 3.2.2. Regulatory Lymphocytes

The immune response against cancer cells is the result of the balance between immune effector cells such as CD8^+^ T cytotoxic cells or NK cells and immunosuppressive cells such as Treg or MDSC [116,117,118].

The resolution of inflammation affects Treg regulation and function(s). It has been demonstrated that SPMs (RvD1, RvD2, and Mar1) exert a regulatory role on the balance between pathogenic Th1 and tolerogenic Treg cells [101,119,120,121] during autoimmune responses. Furthermore, the reduced expression of ALOX15 enzyme in Treg is associated with a lower expression of Foxp3, the master regulator of Treg lineage, and alteration of several cellular metabolic pathways [122]. Consistently, GPR32 plays a crucial role in regulating the adaptive immune responses, as demonstrated by the evidence that its expression is crucial for the generation and function of Treg cells [101]. 

This evidence suggests that the resolution of inflammation could also impact on the role and function of Treg cells in cancer, although several questions remain open. The first point is that the Treg role in cancer initiation/progression is complex: the most accredited theory is that Treg cells exert a pro-tumor effect by inhibiting anti-cancer immune response; however, in some cancer contexts, an anti-tumor effect of Treg was described and associated with their ability to prevent tumor inflammation [123]. Indeed, it has been demonstrated that Treg depletion favors cancer development in a model of inflammation induced colon cancer [43]. Zhang and coll. postulated that Treg could promote early tumor growth, while inhibiting tumor progression later on [123]. Interestingly, they demonstrated that inhibition of Treg in large tumors favors growth, by inducing the generation of MDSCs in the TME, and that this phenomenon is associated with increased levels of LXA4 [123]. 

A specific subpopulation of B cells, namely B regulatory (Breg) cells, have been recently described as able to modulate the immune response in the context of TME [124]. Breg cells are able to produce large amounts of immunosuppressive cytokines (IL-10 and TGF-β) and to express immunoregulatory molecules on their membrane (e.g., PD-L1). Breg cells have been demonstrated to suppress T and NK cells responses in several models of tumor growth [125]. Furthermore, Breg cells facilitate the conversion of CD4^+^ T lymphocyte in TME in CD4^+^CD25^+^FoxP3^+^ Treg cells [125].

Wang and coll. demonstrated that LXA4 is able to suppress the generation of Breg cells in tumor-bearing mice, thus dampening tumor growth. Consistently, the depletion of Breg cells in mice abolishes the anti-tumor activity of LXA4 [53]. Interestingly, LXA4 not only affects Breg cell induction, but is also able to reduce the number of Treg cells, both in the draining lymph nodes and in the TME, and simultaneously to enhance the cytotoxic activities of T lymphocytes [53].

These data suggest that other studies are necessary to clarify the role of SPMs in the modulation of regulatory lymphocytes in the TME, in order to define the effects of SPMs on these immune populations and their impact on cancer progression.

## 4. Effects of SPMs on Cancer-Associated Fibroblasts

Cancer-associated fibroblasts (CAFs) are the most abundant cells in the TME, where they represent a key source of extracellular matrix components, thus contributing to the formation of a desmoplastic stroma and playing crucial roles during malignant cancer progression and metastasis [126].

SPMs promote the regeneration of damaged tissue by finely regulating the fibrotic response, both in the physiology of wound healing and in several pathologic scenarios in which an excessive fibrotic response is established [127]. This is the case of idiopathic pulmonary fibrosis (IPF), in which LXA4 has been reported to inhibit the pro-fibrotic pulmonary response through different mechanisms [128].

The effects of SPMs on CAFs have been studied and described in a few model systems. RvD1 has been demonstrated to inhibit the production of pro-tumorigenic mediators from CAFs in a model of hepatocellular carcinoma (HCC), thus suppressing EMT and the stemness of cancer cells [49]. Furthermore, LXA4 attenuates pancreatic tumor growth by inhibiting the function of human pancreatic stellate cells (hPSCs), which are the precursors of pancreatic CAFs; in more detail, LXA4 attenuates hPSC-induced desmoplasia, reverts hPSC activation into CAFs, and modulates several other hPSC tumor-promoting effects [55]. 

Since one of the problems of the fibrotic process associated with tumors, in addition to the growth-promoting effect, is the formation of a barrier that is difficult to penetrate for drugs, these papers suggest a further potential use of SPMs (i.e., RvD1 and LXA4) as useful agents, in conjunction with conventional therapy, to improve their efficacy.

## 5. Effects of SPMs on the Tumor Vascular Bed

Other than in immune cells, SPM receptors have also been described in endothelial cells (ECs) and vascular smooth muscle cells (VSMCs), and to be modulators of their phenotype and function(s) [13]. In particular, it has been described that ECs express ALX/FPR2, GPR32, GPR18, BLT1, and ChemR23 [16,129,130,131,132]. Among the other types, the anti-angiogenic properties of SPMs have also been described in several contexts [9], for example, mediated by the SPM (LXA4) ability to inhibit EC proliferation and migration, to interfere with vascular endothelial growth factor (VEGF) signaling, and to dampen VEGF receptor expression [133,134]. LXA4 has also been reported to modulate EC functions by increasing NO production [135]. Furthermore, most SPMs affect leukocyte–EC interactions by reducing the expression of adhesion molecules and the production of inflammatory cytokines [13]. Finally, the genetic ablation of GPR32 in ECs demonstrated that the receptor is crucial for endothelial cell integrity and barrier function [136].

However, to date, only a few studies have investigated the role of pro-resolving pathways in the modulation of cancer angiogenesis [9]. 

In a model of Kaposi Sarcoma (KS), LXA4 inhibited the secretion of angiogenic factors [137]. Similarly, LXA4 decreased the production of the angiogenic mediators from HCC cells [138]. Interestingly, SPMs (RvD4 or RvD5), in combination with anti-angiogenic therapy (i.e., the thrombospondin (TSP)-1 peptide 3 TSR or anti-VEGF via DC101), induced synergistic anti-tumor activity in xenograft models of ovarian cancer [138].

Related to cancers of the gastrointestinal (GI) tract, the authors recently described a novel function of SPMs in gastric cancer (GC), demonstrating that two SPMs (i.e., RvD1 and LXB4) suppress angiogenesis, thus inhibiting tumor growth [50]. A key modulator of SPM production was the Formyl Peptide Receptor 1 (FPR1), which functions in the GI tract as a tumor suppressor [139]. FPR1 is a member of an innate immune receptor family of formyl peptide receptors (FPR1, 2, and 3), which are pattern recognition receptors (PRRs) belonging to the G-protein coupled receptor family and able to recognize conserved microbe- or damage-associated molecular structures (PAMPs/DAMPs) and initiate both inflammation and its resolution, depending on the environmental context and on the specific ligand [140,141]. We demonstrated that GC cells silenced for FPR1, but not for FPR2 or FPR3, displayed an in vivo growth advantage, due to their increased ability to form new vessels. In culture, these cells displayed an increase in the production of pro-inflammatory and pro-angiogenic mediators and a drop in the levels of various components of the pro-resolving pathways (ALOX5/15, SPMs (RvD1 and LXB4), and SPM receptors (BLT1, ChemR23 and GPR32)) [50,139]. The same effects were observed when FPR1 was pharmacologically antagonized by cyclosporin H. Consistently, FPR1 activation or increased expression mediates the opposite effects [50]. FPR1-mediated anti-angiogenic potential depends on pro-resolving pathways, as witnessed by the finding that blockade of ALOXs or of SPM receptors (i.e., GPR32) blunted this FPR1 activity in GC cells. Moreover, the increased angiogenic potential of FPR1-depleted GC cells was reverted by exogenous administration of SPMs (RvD1 or LXB4) [50]. The blockade of the pro-resolving receptor GPR32, or of ALOX15, enhanced the angiogenesis and tumorigenic activity of GC cells, mimicking FPR1-depletion. Thus, GC cells are endowed with an intrinsic angiogenic potential, negatively controlled by SPMs. These, in turn, are positively controlled by FPR1. Consistently, a diet enriched in the precursors of SPMs, ω-3 or ω-6 PUFA, [142] inhibited the growth of FPR1-silenced GC cells in mice, by specifically impairing angiogenesis [50].

Several reports have also pointed to a crucial protective role of pro-resolving pathways in colorectal cancer (CRC) [143,144,145]. Unpublished results from the authors’ group also indicate a crucial anti-angiogenic function of SPMs in CRC: the activation of FPR1 mediated by the probiotic *Lactobacillus Rhamnosus* (LGG) or specific bacterial products in CRC cells caused an increase of pro-resolving mediators (RvD1 and LXB4) and the consequent inhibition of angiogenesis [51]. *LGG* is a commensal bacterium used as a probiotic and described as able to dampen the chronic inflammation associated with CRC development [146]. *LGG* sustains colonic wound healing in mice [147,148,149] and activates pro-apoptotic and anti-metastatic responses [150,151] through the interaction with FPR1 [147]. Our data highlight new pro-resolving and anti-angiogenic functions of this bacterium mediated by FPR1 [51].

By asking whether similar mechanisms could also intervene in cancers derived from other epithelia, we demonstrated that pro-resolving pathways could also exert anti-angiogenic activity in a lung cancer model; however, in this context, the innate immune receptor that controls SPM production is the Toll-like receptor 7 (TLR7) [52]. In this model system, TLR7 activation, similarly to FPR1 in the GI tract, sustains the increased expression of the enzymes responsible for SPMs biosynthesis, the increased secretion of SPMs from cancer cells, the augmented expression of SPM receptors, and finally the reduction of the angiogenic potential of lung cancer cells [52].

## 6. Conclusions

Inflammation has long been considered a hallmark of cancer, both for its ability to intervene in cancer initiation and to sustain cancer progression. Recently, the discovery of the resolution of inflammation opened the way for new therapeutic possibilities that, rather than block the inflammatory response, could instead support the induction of resolution of the inflammation.

Among others, Specialized Pro-resolving Mediators (SPMs) are “bioactive lipids” already described as key players in regulating inflammatory response and tissue homeostasis in several physiologic and pathologic conditions, by acting on several cell types in the context of inflammatory reactions. The present review focused on the possibility that SPMs could shape the TME, to affect cancer cell growth and progression.

Strong evidence has been provided supporting the possibility that SPMs act on the innate immune cell compartment infiltrating tumors from an anti-cancer perspective (Figure 1).

Indeed, SPMs favor the reduction of the inflammatory response and the acquisition of a “pro-resolving” phenotype in these cells, which contributes to the anti-tumor response. Less data are available on the effects of SPMs on the adaptive immune compartment, and no clear definition of the function of SPMs in the context of an adaptive anti-cancer response has been defined (Figure 1).

Interestingly, SPMs have been described to act on various other cells constituting tumor stroma (Figure 1). Consistently with their natural anti-fibrotic and anti-angiogenic properties, SPMs have been described as able to counteract the CAF ability to sustain cancer progression and to dampen the angiogenic response vital for cancer dissemination.

Although further studies are needed to define the role of these bioactive lipids in cancer, they appear to be a potential and promising approach to fight cancer progression.

## Figures and Tables

**Figure 1 cancers-14-03333-f001:**
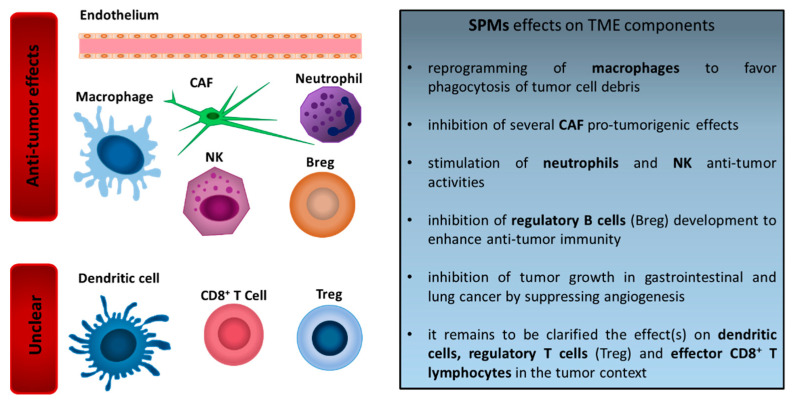
Effect of SPMs on TME components.

**Table 1 cancers-14-03333-t001:** Cancer models and the mechanisms by which specific SPMs inhibit cancer progression by directly acting on tumor cells or by targeting different components of the TME.

	SPM	Mechanism(s)	Cancer Model	References
Direct effects on cancer cells	LXA4	suppression of cancer cell growth and invasion	lung	[34]
		suppression of cancer cell growth and invasion	hepatocellular carcinoma	[41]
		reduction of tumor cell migration	acute cell leukaemia	[41]
		anti-estrogenic activity	endometrium	[40]
	RvD1	inhibition of cancer cell proliferation	oral	[42]
		inhibition of epithelial to mesenchymal transition	lung	[39]
Effects on TME	RvD1	targeting of regulatory cells	colon	[43]
		clearance of cell debris by macrophage	pancreas, lung	[44,45]
		modulation of neutrophil phenotype and recruitment of anti-cancer monocytes	papilloma	[46]
		modulation of TAM phenotype	prostate	[47]
		increase of NK function	pancreas	[38]
		reduction neutrophil infiltration	lung	[48]
		inhibition of CAF pro-tumorigenic mediators	HCC	[49]
		reduction of angiogenic response	stomach, colon, lung	[50,51,52]
	RvD2	clearance of cell debris by macrophage	pancreas, lung	[44,45]
		modulation of TAM phenotype	prostate	[47]
	RvE1	clearance of cell debris by macrophage	pancreas, lung	[44,45]
	LXA4	targeting Breg cells and increasing cytotoxic T cell activity	liver	[53]
		modulation of TAM phenotype	melanoma	[54]
		inhibition of CAF precursors	pancreas	[55]
	LXB4	reduction of angiogenic response	stomach, colon, lung	[50,51,52]

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
