# Peer review of "The Impact of Resolution of Inflammation on Tumor Microenvironment: Exploring New Ways to Control Cancer Progression"

_cancers, 2022, doi:10.3390/cancers14143333_

Round 1

Reviewer 1 Report

The review article by Liotti et al., is well written and should be of interest to those involved in cancer biology and therapeutics, however, there are some areas for improvement which will make for a more balanced review. 

As with many reviews on SPMs, the receptors that these lipids activate are treated as fully validated and are typically included as a single summary sentence (see line 90-92). Unfortunately, there is a growing body of work that cannot replicate many of the original pairings (pmid: 35125345, pmid: 27018161), and such the mechanism by which these SPMs exert their effects still requires rigorous validation, therefore, 

·       a section describing what is known about the expression and function of these receptors within the TME and how they might be affected differentially between cell types and cancer should be included.

I was interested to read lines 371-386 where the authors describe the role of FPR1 on SPM expression and activity. To date FPR1 has not been widely ascribed to SPM biology – this is mostly driven by FPR2 – and FPR1 is not listed as an SPM receptor by the authors in lines 90-92. Are the authors suggesting that FPR1 is a key regulator of these SPMs and their receptors? Could these effects be attributed to redundancy within the FPR family?

·      A summary prior to this section describing what functional data is published for activation of FPR1 by SPMs should be included. 

·      More detail should be given explaining functionally how deletion/ablation results in a decrease in “(RvD1 and LXB4), and SPM receptors (BLT-1, ChemR23 and GPR32)], and the consequent increase in the production of pro-inflammatory and pro-angiogenic mediators”

·     A summary of the other SPM receptors (lines 90-92) on the tumor vascular bed should be included rather than solely focusing on FPR1.

Author Response

The review article by Liotti et al., is well written and should be of interest to those involved in cancer biology and therapeutics, however, there are some areas for improvement which will make for a more balanced review. 

 As with many reviews on SPMs, the receptors that these lipids activate are treated as fully validated and are typically included as a single summary sentence (see line 90-92). Unfortunately, there is a growing body of work that cannot replicate many of the original pairings (pmid: 35125345, pmid: 27018161), and such the mechanism by which these SPMs exert their effects still requires rigorous validation, therefore,

  • a section describing what is known about the expression and function of these receptors within the TME and how they might be affected differentially between cell types and cancer should be included.

We thank the reviewer for the comment and the suggestion. We agreed and added more information better describing SPM receptors specificity and the discussed data available in the literature, including that indicated by the reviewer (please see lines 93-106 and references no. 23 and 24).

In particular, to allow the discussion of the receptor function(s) in different cells of the TME, we decided to describe the data available in the literature including a section regarding the SPM receptors in the paragraph of each cell population. Please see lines 198-204 (neutrophils), lines 243-250 (macrophages), 339-344 (lymphocytes), lines 365-367 (Treg).

I was interested to read lines 371-386 where the authors describe the role of FPR1 on SPM expression and activity. To date FPR1 has not been widely ascribed to SPM biology – this is mostly driven by FPR2 – and FPR1 is not listed as an SPM receptor by the authors in lines 90-92. Are the authors suggesting that FPR1 is a key regulator of these SPMs and their receptors? Could these effects be attributed to redundancy within the FPR family?

Thanks for the comment. As underlined by the reviewer, we did not provide data showing that FPR1 is the receptor for any SPM. Rather, in our previous work, we showed that FPR1, when stimulated by its known ligands, is a key activator of the pro-resolving activity in GC (and CRC cells). In fact, FPR1 stimulation promotes the expression of ALOX5/15 by increasing the relative mRNAs. This results in an increase of SPM production. Moreover, by increasing the transcription of SPM receptors, FPR1 renders GC cells highly responsive to SPM stimulation.

As far as redundancy is concerned, we clearly demonstrated, by using both genetic and pharmacologic approaches, that FPR1, but not FPR2 or FPR3, could control SPM production. Thus, at least in GC cells, inflammation resolution seems to be specifically controlled by FPR1.

  • A summary prior to this section describing what functional data is published for activation of FPR1 by SPMs should be included.

As mentioned above, FPR1 is not described to date as receptor for SPMs. Thus, we did not include FPR1 in the section describing receptors for SPMs.

  • More detail should be given explaining functionally how deletion/ablation results in a decrease in “(RvD1 and LXB4), and SPM receptors (BLT-1, ChemR23 and GPR32)], and the consequent increase in the production of pro-inflammatory and pro-angiogenic mediators”

Done, please see line 447-461.

  • A summary of the other SPM receptors (lines 90-92) on the tumor vascular bed should be included rather than solely focusing on FPR1.

Thanks for the suggestion, we included a paragraph better describing the effects of other SPMs on the vascular bed and, when available, data on receptor involvement. Please see lines 419-430

Reviewer 2 Report

This is a very detailed review. kindly cite

(1) krishnamoorthy et al J Immunol. 2015 Feb

(2)De Matteis R et al Sci Adv. 2022 Feb

(3) include how SPMs can counteract  immune checkpoints markers (CTLA-4 and PD-1

Author Response

This is a very detailed review. kindly cite

(1) krishnamoorthy et al J Immunol. 2015 Feb

(2)De Matteis R et al Sci Adv. 2022 Feb

The two references were added to the new version of the manuscript. Please see no.115 and 61

(3) include how SPMs can counteract  immune checkpoints markers (CTLA-4 and PD-1)

Many thanks for the suggestion. You will find a section dedicated to SPM effects on immune checkpoint at lines 149-176.